# Exploring COVID-19 vaccine uptake among healthcare workers in Zimbabwe: A mixed methods study

Tinotenda Taruvinga[1,2,3]*, Rudo S. Chingono[1], Edson Marambire[1,4], Leyla Larsson[1,4], Ioana D. Olaru[1,5], Sibusisiwe Sibanda[1], Farirai Nzvere[1], Nicole Redzo[1], Chiratidzo E. Ndhlovu[6], Simbarashe Rusakaniko[7], Hilda Mujuru[8], Edwin Sibanda[9], Prosper Chonzi[10], Maphios Siamuchembu[11], Rudo Chikodzore[12], Agnes Mahomva[13], Rashida A. Ferrand[1,5], Justin Dixon[1,2]☯, Katharina Kranzer[1,4,5]☯

1 The Health Research Unit Zimbabwe, Biomedical Research & Training Institute, Harare, Zimbabwe, 2 Department of Global Health and Development, London School of Hygiene and Tropical Medicine, London, United Kingdom, 3 Africa Centres for Diseases Prevention and Control (Africa CDC), Addis Ababa, Ethiopia, 4 Division of Infectious Diseases and Tropical Medicine, LMU University Hospital, LMU Munich, Germany, 5 Clinical Research Department, London School of Hygiene and Tropical Medicine, London, United Kingdom, 6 Internal Medicine Unit, University of Zimbabwe Faculty of Medicine and Health Sciences, Harare, Zimbabwe, 7 Department of Community Medicine, College of Health Sciences, University of Zimbabwe, Harare, Zimbabwe, 8 Department of Paediatrics and Child Health, University of Zimbabwe College of Health Sciences, Harare, Zimbabwe, 9 Bulawayo City Council Health Department, Bulawayo, Zimbabwe, 10 Harare City Council Health Department, Harare, Zimbabwe, 11 Ministry of Health and Child Care, Provincial Medical Directorate, Bulawayo, Zimbabwe, 12 Ministry of Health and Child Care, Department of Epidemiology and Diseases Control, Harare, Zimbabwe, 13 National Response to the COVID-19 Pandemic, Office of the President, and Cabinet, Harare, Zimbabwe

☯ These authors contributed equally to this work.
* Tinotenda.Taruvinga1@lshtm.ac.uk

**Data Availability Statement:** Data are available open access via London School of Hygiene and

## Abstract

With COVID-19 no longer categorized as a public health emergency of international concern, vaccination strategies and priority groups for vaccination have evolved. Africa Centres for Diseases Prevention and Control proposed the '100-100-70%' strategy which aims to vaccinate all healthcare workers, all vulnerable groups, and 70% of the general population. Understanding whether healthcare workers were reached during previous vaccination campaigns and what can be done to address concerns, anxieties, and other influences on vaccine uptake, will be important to optimally plan how to achieve these ambitious targets. In this mixed-methods study, between June 2021 and July 2022 a quantitative survey was conducted with healthcare workers accessing a comprehensive health check in Zimbabwe to determine whether and, if so, when they had received a COVID-19 vaccine. Healthcare workers were categorized as those who had received the vaccine 'early' (before 30.06.2021) and those who had received it 'late' (after 30.06.2021). In addition, 17 in-depth interviews were conducted to understand perceptions and beliefs about COVID-19 vaccines. Of the 3,086 healthcare workers employed at 43 facilities who participated in the study, 2,986 (97%, 95% CI [92%-100%]) reported that they had received at least one vaccine dose. Geographical location, older age, higher educational attainment and having a chronic condition was associated with receiving the vaccine early. Qualitatively, (mis)information, infection risk perception, quasi-mandatory vaccination requirements, and legitimate

Tropical Medicine (LSHTM) Data Compass https://doi.org/10.17037/DATA.00003679.

**Funding:** This work was supported by the Global Public Health strand of the Elizabeth Blackwell Institute for Health Research, funded under the University of Bristol's QR GCRF strategy (ref: H100004-148) as well as support from Sheffield and Oxford QR-GCRF funds. It was supported by UK aid from the UK government (FCDO) (ref 668 303), and by funding from the government of Canada. The views expressed do not necessarily reflect the policies of the respective governments. RAF is funded by a Wellcome Trust Senior Fellowship (206316_Z_17_Z). IDO has received funding through the Wellcome Trust Clinical PhD Programme awarded to the London School of Hygiene & Tropical Medicine (grant number 203905/Z/16/Z). The funders had no role in study design, data collection and analysis, decision to publish, or preparation of the manuscript. Tinotenda Taruvinga is supported by the Fogarty International Centre of the National Institutes of Health (NIM; Bethesda, Maryland, MD, USA) under Award Number D43 TW009539. The content therein is solely the responsibility of the authors and does not necessarily represent the official views of the National Institutes of Health.

**Competing interests:** The authors have declared that no competing interests exist.

concerns such as safety and efficacy influenced vaccine uptake. Meeting the proposed 100-100-70 target entails continued emphasis on strong communication while engaging meaningfully with healthcare workers' concerns. Mandatory vaccination may undermine trust and should not be a substitute for sustained engagement.

## Introduction

Coronavirus disease 2019 (COVID-19) vaccines have been a key pillar of the pandemic response at global, national, and local levels. Their roll-out has reduced morbidity, severity and deaths as reported in many vaccine effectiveness studies [1–3]. However, the hoarding of vaccines by wealthier countries (i.e. 'vaccine nationalism') and global unequal vaccine distribution limited the availability of COVID-19 vaccines, especially in low- and middle-income countries (LMICs) [4, 5], resulting in slow and often erratic roll out [5]. In response to vaccine nationalism and accessibility challenges, various international platforms were created to increase vaccine availability in LMICs, including the COVID-19 vaccine delivery partnership (CoVDP) (which is an alliance of the WHO, UNICEF and GAVI) and the COVID-19 vaccines global access (COVAX) platform [4, 6]. While many LMICs signed up to the COVAX initiative, others opted for bilateral arrangements [7]. For instance, Zimbabwe the focus of our research, did not initially sign up to the COVAX initiative and instead obtained vaccines through bilateral arrangements with China, Russia and India [4, 7].

To optimize the use of limited and often unpredictable supply of vaccines, Zimbabwe like many LMICs used a phased approach that prioritized at-risk groups for vaccination, including healthcare workers [8, 9]. Prioritizing and ensuring high vaccine uptake among healthcare workers was important for several reasons. First, healthcare workers were widely recognized as being at heightened risk of COVID-19, as reflected in the high mortality rate during the pandemic [10, 11]. Second, vaccination among healthcare workers has further ramifications in terms of preventing nosocomial transmission and more broadly for ensuring the human resourcing of health systems [12, 13]. Third, vaccine uptake among healthcare workers has a considerable influence on uptake among the general population [9, 13]. Studies from both high-income countries and LMICs suggest a considerable proportion of the general population would consider healthcare workers' advice before vaccination [14–16].

Vaccine strategies have evolved over the last two years and new recommendations on who should be prioritized for COVID-19 vaccination have been released. On the 5th of May 2023, the WHO has de-escalated the COVID-19 pandemic as a public health emergency of international concern [17]. Related to the new recommendations, Africa CDC in their new strategy have maintained healthcare workers as a priority group with a target set to reach 100% (timeline for target unspecified) COVID-19 vaccine coverage among this group [18]. In view of this ambitious target of vaccinating all healthcare workers, it is crucial to understand whether past COVID-19 vaccination strategies were successful and what could be done to address healthcare workers' perceptions, anxieties, and concerns to reach the ambitious 100% target. Strategies to achieve universal vaccine coverage will likely need to be context-sensitive and informed by actual uptake data during the pandemic. In this mixed-methods study, we sought to understand vaccine uptake by describing the 'early' and 'late' uptake, including qualitatively exploring the perceptions, and attitudes among healthcare workers in Zimbabwe.

## Methods

### Study design

Data was collected as part of a broader study providing a comprehensive health check to healthcare workers in Zimbabwe which has been described in detail elsewhere [19]. A quantitative survey and in-depth interviews were conducted with selected healthcare workers accessing the comprehensive health check service between June 2021 and July 2022.

### Study setting and population

Zimbabwe is a low-income country with a long history of severe economic decline affecting healthcare services, public health programmes, and epidemic management capacity [20, 21]. During the study period, healthcare workers had taken up industrial action over low wages and unavailability of adequate personal protective and medical equipment [20, 22]. The study was conducted in public hospitals across all ten provinces in Zimbabwe and primary care clinics in Harare, Matabeleland North, and Mashonaland East. The study participants were clinical and non-clinical healthcare workers who accessed the comprehensive health check service.

### Zimbabwe's COVID-19 vaccination campaign

Zimbabwe launched its COVID-19 vaccination campaign on 22 February 2021 and started distributing booster doses in December 2021 [23, 24]. The country approved the use of Sinopharm and Sinovac vaccines before they had received Emergency Authorization Use (EAU) by the WHO [25], followed by Covax, Sputnik V and Johnson and Johnson [26]. However, only Sinopharm and Sinovac were readily available in the public sector [27].

Vaccines were supplied in batches and were initially distributed primarily in Harare and Bulawayo (the two largest cities in Zimbabwe) which had reported the highest number of SARS-CoV-2 infections. Other provinces received vaccines as the supply chain improved. By 31 July 2022, an estimated 11,8 million doses had been administered, resulting in a target population coverage of 53.2% for the first dose, 41.3% for the second dose, and 7.4% for the third dose [28]. Fig 1 shows administered vaccine doses and SARS-CoV-2 infections over time. The highest number of vaccine doses were administered during the COVID-19 pandemic third wave, which was primarily driven by the more contagious SARS-CoV-2 delta variant and resulted in the highest number of deaths.

### Procedures

Prior to accessing the health check services, verbal informed consent was obtained. Data on age, sex, professional role, education, clinical history, past SARS-CoV-2 infection and COVID-19 vaccination history, and perceived vaccine safety were obtained from all the healthcare workers accessing the service. The quantitative component involved a questionnaire based on a WHO survey tool [18] which asked about COVID-19 vaccinations history and perceived vaccine safety in addition to the already provided socio demographics and clinical history data. Subsequent to the questionnaire and health check, written informed consent was obtained from a sub-set of healthcare workers purposively (i.e. identifying participants sharing particular characteristics not seeking a representative sample but rather a non-probability sample focused on identifying participants fitting the objectives of the study, [29]) selected for in-depth interviews [30] based on the vaccination status, the time of their first vaccination dose and the place they worked. The in-depth interviews were administered using a topic guide. This was to better understand healthcare workers reasons for taking or not taking up the COVID-19 vaccine and the challenges they may have faced in the process.

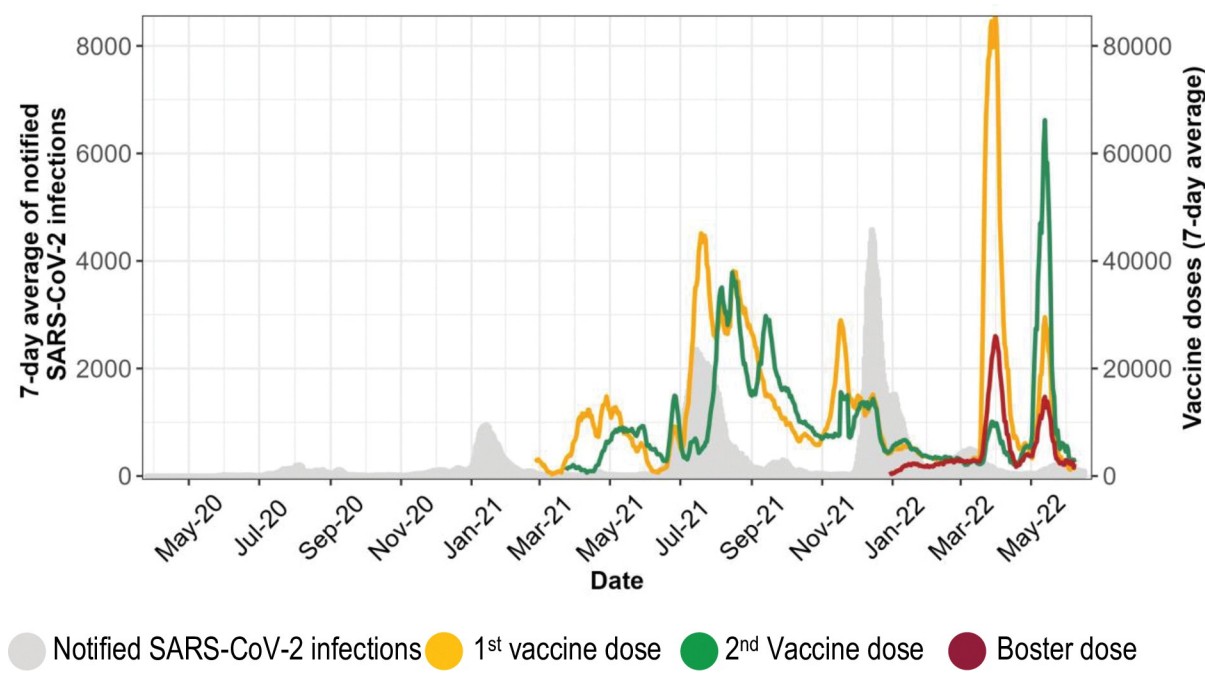

**Fig 1. Routine data on vaccine doses administered and notified SARS-CoV-2 infections obtained from the daily published Ministry of Health and Child Care situational reports.**

A total of 17 in-depth interviews were conducted, after reaching a data saturation point (i.e. when we were no longer generating new information to existing categories) [30]. A topic guide was developed prior to the interview and included questions about participants' vaccination status, challenges of accessing vaccines, concerns about vaccine safety, reasons for being vaccinated or not, and sources of information to guide decision-making. During interviews, participants were given a broader remit to discuss more general concerns and anxieties, as well as why there were these concerns in context, including what specifically concerned them as healthcare workers. While the questions asked as part of the quantitative questionnaire specifically asked about personal reasons for not getting vaccinated (if they had not been vaccinated), the in-depth interview guide gave healthcare workers room to express both positive and negative anxieties they had and those their patients, families and communities may have voiced. Both the questionnaire and interviews were conducted in English, Ndebele, or Shona according to the participant's choice. Interviews lasted between 30 to 75 minutes while interviewing stopped after exhausting all possible probing questions (i.e., questions that enable you to follow up on areas of interest related to an overarching theme, some of which might not be predictable in advance [30]). This was done to explore, in greater detail respondents' perceptions and feelings about taking up or not taking up the COVID-19 vaccines.

## Data management and analysis

Quantitative data were collected using electronic real time data entry (SurveyCTO). Electronic questionnaires were uploaded daily and saved to a Microsoft SQL server. Participants were identified by a unique identification number. The data was analysed in R version 4.1.2. Means with associated standard deviations (SD) were computed for normally distributed continuous variables; medians and interquartile ranges (IQR) were analysed for non-normality distributed continuous variables. Absolute counts and proportions for categorical variables were

calculated. The main outcome variable was receiving the vaccine 'early' or 'late'. Healthcare workers categorised as receiving the vaccine "early" were defined as having received the first vaccine dose between 22nd of February and 30th of June 2021, while those receiving the first vaccine dose after June 2021 were categorised as "late". Those who did not have a response to the vaccination variable were excluded from analysis.

A multivariable logistic regression was conducted to investigate associations between client characteristics (sex, age, occupation, comorbidities, province of origin, administrative authority, and prior history of SARS-CoV-2 infection) and receiving the vaccine "late". Variables included in the final model were purposefully selected and analysis was conducted on a complete observation set. P-values were derived using a likelihood ratio test. Results are reported as adjusted odds ratios (aOR) along with their corresponding 95% confidence intervals (95% CI). Those who did not take the vaccine were excluded from the main regression analysis but were however included in the "late" vaccination group in a sensitivity analysis (whose results we have shown in S1 Table).

Staff roles were categorised into clinical and non-clinical, where clinical staff included, nurses, doctors, laboratory technicians, nurse aides, radiographers, rehabilitation technicians, and pharmacists. Non-clinical staff included administrators, environmental health practitioners, security guards, cleaners, and health information staff. Health facilities were categorised based on their level of care: primary (polyclinics), secondary (district and mission hospitals), tertiary (provincial hospitals) and quaternary (central specialised group of hospitals). These facilities were either owned by local authority, the central government, or faith-based organisations. Past medical history of co-morbidities was coded as i) none ii) one or iii) two or more co-morbidities. Co-morbidities which were ascertained through self-report included: HIV, asthma, chronic lung disease, cancer, diabetes, hypertension, cardiovascular disease, and chronic kidney disease. Body mass index (BMI) was calculated using weight and height. Healthcare workers with a BMI <18.5, 18.5–24.9, 25–29.9, and >30 were categorised as underweight, healthy, overweight, and obese respectively.

In-depth interviews were audio-recorded, transcribed, and translated by trained research assistants. During the interview, research assistants took field notes and wrote interview summaries at the end of the day. Transcripts and other qualitative data (i.e., field notes and interview summaries) were imported into the qualitative data analysis software NVivo 12. Within an overarching grounded theory approach, [30] we used thematic analysis [29] that ran synchronously with, and subsequent to, the collection of data. Interim analysis of an initial set of transcripts to identify patterns within the data, based on which initial themes and working hypotheses were generated. Hypotheses generated from this interim analysis were fed back into the topic guide and subsequent interviews to refine and, where appropriate, change our working hypotheses. Working iteratively back and forth between analysis and data collection, we worked towards progressively broader and more encompassing themes to explain and theorise findings. Quantitative findings were triangulated with the themes emerging from the qualitative data throughout the analysis process to create meaning of the findings. The overall aim of this bottom-up analytic process was to produce a robust, contextual understanding and interpretation of the vaccine uptake among healthcare workers in Zimbabwe.

## Ethical approval

Ethical approval was obtained from the Medical Research Council Zimbabwe (MRCZ/A/2627); the Biomedical Research and Training Institute and the London School of Hygiene and Tropical Medicine ethics committees (22514). For healthcare workers accessing the health check service and responding to the quantitative questionnaire, verbal informed consent was

obtained. The Medical Research Council Zimbabwe waived the necessity for a written informed consent to facilitate access to the service. Written informed consent was obtained from all participants for participation in the in-depth interviews.

### Inclusivity in global research

Additional information regarding the ethical, cultural, and scientific considerations specific to inclusivity in global research is included in S1 Checklist.

## Results

### Reported vaccine uptake among health care workers

A total of 3,086 healthcare workers from 43 health facilities accessed the service during the study period, half of which worked at facilities in Harare and Bulawayo provinces (Table 1). Three-quarters (2,339/3,086, 75.8%) were women, and median age was 37 (IQR: 28–46) years. The majority of healthcare workers had clinical roles (2,979/3,086, 96.5%). A total of 931 (30.2%) healthcare workers had one or more known chronic diseases, and 64.1% (1,978/3,086) were either overweight or obese and about a third (1,086/3,086, 35.2%) worked at a government-owned institution. Past SARS-CoV-2 infections were reported by 1,119 (36.3%) healthcare workers.

Almost all healthcare workers (2,986/3,086, 96.8%) reported having received at least one dose of COVID-19 vaccine, and most had received two doses (2,905/2,986, 97.3%). Half of the healthcare workers received their first dose before the 30th of June 2021 (1,535/2,986, 51.4%). Sinopharm (1,747/2,986, 58.5%) was the most frequently administered vaccine followed by Sinovac (1,059/2,986, 35.5%); the remaining 2% had received Sputnik V or were not sure of the vaccine they had received.

In the multivariable model, age, number of years in current role, administrative authority, and province were associated with receiving the vaccine late. Healthcare workers of older age (aOR: 0.48, 95% CI: 0.37–0.62) were less likely to receive the vaccine late. Clients who were working in their current role for less than one year (aOR: 4.01, 95% CI: 3.12–5.14) and those working at a mission or private institution (aOR: 2.32, 95% CI: 1.54–3.49), however, were more likely to receive the vaccine late. There was evidence of association between province where the health facility was located and time of vaccine uptake, though it varied by province (Table 2).

### Factors influencing vaccine uptake

Please note we have included appropriate themes, codes, and additional quotes in S2 Table.

**(Mis)Information.** Participants generally felt that the government had run an effective information campaign using radio, television, billboards, and banners. The information provided, they felt, was reliable, accurate, and easy to understand, and included messages encouraging the population to get vaccinated. However, it was highlighted that information was not always communicated in all languages, which impacted the accessibility of information.

> "*In terms of information I think they have done pretty well, maybe they will just put in all languages because at times you would go to get news and you see it will be written in Shona only, there is no Ndebele poster. It's not everyone who will understand what they are saying. So, they should just try and put it in all languages so that everyone in their respective places they read and understand.*" (Clinician, Late Receiver, Matabeleland North).

**Table 1. Baseline characteristics of healthcare workers accessing the occupational health services stratified by time of COVID-19 vaccination.**

| Variables | | Early vaccination (Feb 21—June 21) N (%) | Late vaccination (After June 21) N (%) | Non-vaccinated, N (%) |
|---|---|---|---|---|
| | Total | 1,535 | 1,451 | 100 |
| **Sex** | Male (n = 745) | 382 (51.3) | 342 (45.9) | 21 (2.8) |
| | Female (n = 2,339) | 1,152 (49.3) | 1,108 (47.4) | 79 (3.4) |
| **Age (years)** | < 30 (n = 944) | 321 (34.0) | 573 (60.7) | 50 (5.3) |
| | 30–40 (n = 914) | 453 (49.6) | 428 (46.8) | 33 (3.6) |
| | > 40 (n = 1,226) | 760 (62.0) | 449 (36.6) | 17 (1.4) |
| **Role** | Clinical (n = 2,979) | 1,478 (49.6) | 1,402 (47.1) | 99 (3.3) |
| | Non-clinical (n = 105) | 56 (53.3) | 48 (45.7) | 1 (1.0) |
| **Years at current role** | < 1 (n = 574) | 113 (19.7) | 426 (74.2) | 35 (6.1) |
| | 1–5 (n = 1,225) | 646 (52.7) | 544 (44.4) | 35 (2.9) |
| | 6–10 (n = 478) | 284 (59.4) | 183 (38.3) | 11 (2.3) |
| | > 10 (n = 807) | 491 (60.8) | 297 (36.8) | 19 (2.4) |
| **Number of known comorbidities** | None (n = 2,153) | 977 (45.4) | 1,101 (51.1) | 75 (3.5) |
| | 1 (n = 770) | 454 (59.0) | 297 (38.6) | 19 (2.5) |
| | > 1 (n = 161) | 103 (64.0) | 52 (32.3) | 6 (3.7) |
| **History of SARS-CoV-2 infection** | Yes (n = 1,119) | 587 (52.5) | 503 (45.0) | 29 (2.6) |
| | No (n = 1,967) | 948 (48.2) | 948 (48.2) | 71 (3.6) |
| **Body mass index** | Underweight (n = 82) | 33 (40.2) | 46 (56.1) | 3 (3.7) |
| | Healthy (n = 1,015) | 452 (44.5) | 517 (50.9) | 46 (4.5) |
| | Overweight (n = 930) | 475 (51.1) | 434 (46.7) | 21 (2.3) |
| | Obese (n = 1,048) | 570 (54.4) | 448 (42.7) | 30 (2.9) |
| **Highest level of education** | O-levels (n = 1,293) | 625 (48.3) | 618 (47.8) | 50 (3.9) |
| | A-levels (n = 362) | 138 (38.1) | 213 (58.8) | 11 (3.0) |
| | Diploma (n = 1,057) | 569 (53.8) | 454 (43.0) | 34 (3.2) |
| | University (n = 372) | 202 (54.3) | 165 (44.4) | 5 (1.3) |
| **Type of facility** | Primary (n = 950) | 528 (55.6) | 380 (40.0) | 42 (4.4) |
| | Secondary (n = 1,187) | 567 (47.8) | 593 (50.0) | 27 (2.3) |
| | Tertiary (n = 770) | 323 (41.9) | 421 (54.7) | 26 (3.4) |
| | Quaternary (n = 155) | 98 (63.2) | 52 (33.5) | 5 (3.2) |
| **Administrative authority** | Local (n = 1,603) | 878 (54.8) | 670 (41.8) | 55 (3.4) |
| | Government (n = 1,086) | 481 (44.3) | 569 (52.4) | 36 (3.3) |
| | Mission or private (n = 391) | 172 (44.0) | 210 (53.7) | 9 (2.3) |
| **Province** | Harare (n = 959) | 536 (55.9) | 381 (39.7) | 42 (4.4) |
| | Bulawayo (n = 467) | 244 (52.2) | 211 (45.2) | 12 (2.6) |
| | Mashonaland East (n = 271) | 81 (29.9) | 182 (67.2) | 8 (3.0) |
| | Mashonaland West (n = 275) | 117 (42.5) | 144 (52.4) | 14 (5.1) |
| | Mashonaland Central (n = 161) | 60 (37.3) | 96 (59.6) | 5 (3.1) |
| | Masvingo (n = 165) | 60 (36.4) | 98 (59.4) | 7 (4.2) |
| | Manicaland (n = 158) | 91 (57.6) | 65 (41.1) | 2 (1.3) |
| | Midlands (n = 164) | 71 (43.3) | 90 (54.9) | 3 (1.8) |
| | Matabeleland South (n = 331) | 184 (55.6) | 142 (42.9) | 5 (1.5) |

⁑ = Row percentages

**Table 2. Multivariable logistic regression investigating the association between client characteristics and receiving the vaccine late (n = 2,986).**

| Variables | | aOR | 95% CI | p-value |
|---|---|---|---|---|
| **Sex** | Male | - | | 0.49 |
| | Female | 1.07 | 0.88–1.30 | |
| **Age (years)** | < 30 | - | | < 0.001 |
| | 30–40 | 0.74 | 0.58–0.94 | |
| | > 40 | 0.48 | 0.37–0.62 | |
| **Years at current role** | < 1 | 4.01 | 3.12–5.14 | < 0.001 |
| | 1–5 | - | | |
| | 6–10 | 0.94 | 0.74–1.20 | |
| | > 10 | 0.99 | 0.79–1.24 | |
| **History of SARS-CoV-2 infection** | Yes | 0.80 | 0.50–1.29 | 0.86 |
| | No | - | | |
| **Body mass index** | Underweight | 1.00 | 0.60–1.65 | 0.93 |
| | Healthy | - | | |
| | Overweight | 1.03 | 0.84–1.27 | |
| | Obese | 1.01 | 0.81–1.25 | |
| **Highest level of education** | O-levels | - | | 0.09 |
| | A-levels | 0.92 | 0.69–1.22 | |
| | Diploma | 0.86 | 0.71–1.03 | |
| | University | 0.81 | 0.63–1.05 | |
| **Administrative authority** | Local | - | | < 0.001 |
| | Government | 0.85 | 0.63–1.15 | |
| | Mission or private | 2.32 | 1.54–3.49 | |
| **Province** | Harare | - | | < 0.001 |
| | Bulawayo | 0.89 | 0.68–1.16 | |
| | Mashonaland East | 3.21 | 2.10–4.91 | |
| | Mashonaland West | 1.75 | 1.25–2.46 | |
| | Mashonaland Central | 2.79 | 1.74–4.46 | |
| | Masvingo | 0.93 | 0.54–1.61 | |
| | Manicaland | 1.04 | 0.65–1.67 | |
| | Midlands | 1.33 | 0.83–2.14 | |
| | Matabeleland South | 0.59 | 0.41–0.85 | |

p-value was derived using a likelihood ratio test.

Abbreviations: aOR: adjusted odds ratio, 95% CI: 95% confidence interval

Additionally, information about accessing the vaccine while breastfeeding, during pregnancy, and if HIV-positive was not consistent, especially at the start of the vaccination campaign.

> "*I think the message wasn't so clear [from the government], people had so many questions about the vaccine. Like who is eligible for the vaccines, for example at first it was said pregnant women are not supposed to be vaccinated. Like as healthcare workers you are just telling people that pregnant women should not be vaccinated but after some time, they say pregnant women are eligible. Already people are having some sort of confusion and others are still holding on to that old information. And I think that information they gave us it wasn't really enough.*" (Clinician, Early Receiver, Bulawayo)

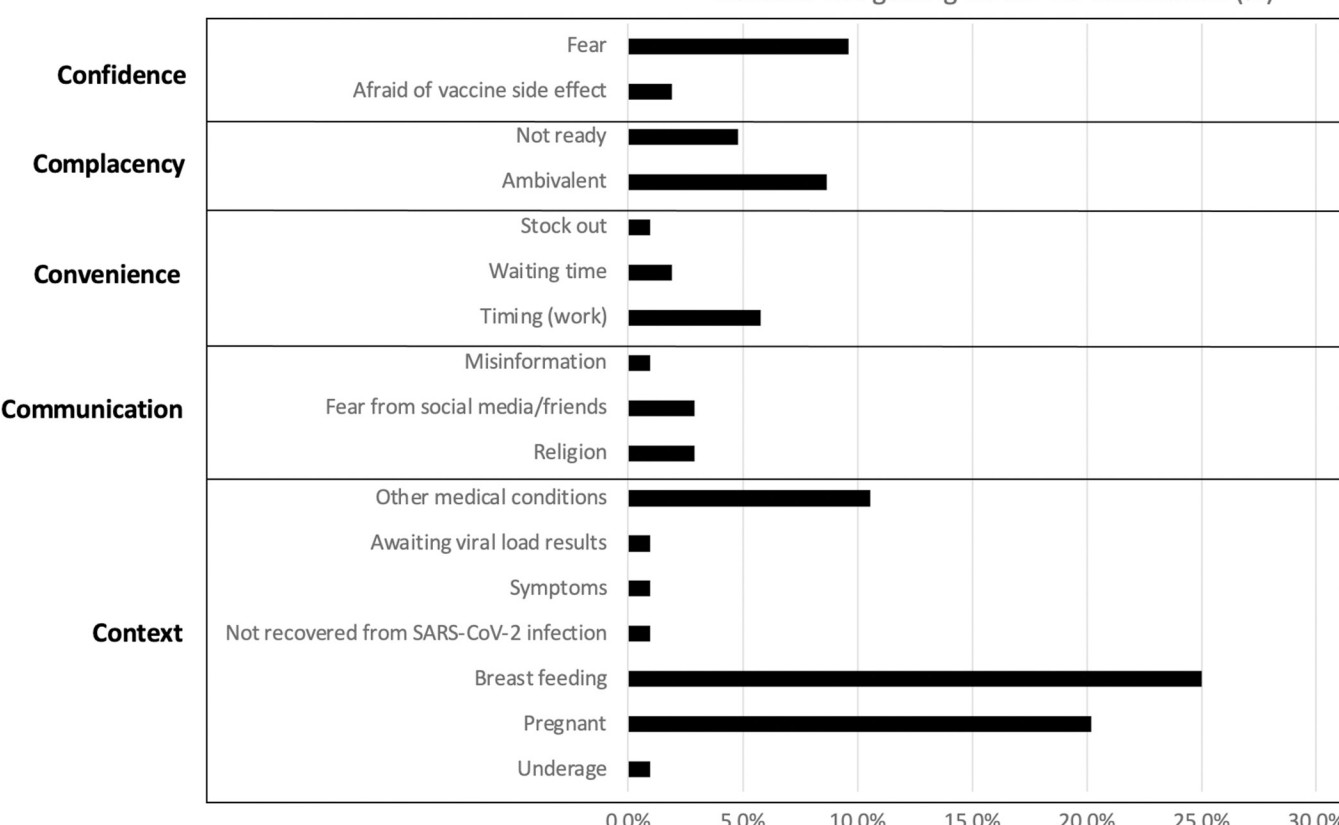

**Fig 2. Bar graph showing reasons for no COVID-19 vaccination among healthcare workers (N = 100) based on the 5Cs of hesitancy model [31].**

This was supported by the quantitative data (Fig 2) showing that the most frequently reported reason for not being vaccinated was pregnancy, breastfeeding or trying to conceive (47/98, 48%), followed by fear of side effects (12/98, 12.2%).

While some participants felt that the government had provided reliable information about the vaccines, most participants reported the internet and social media being their main sources of information. However, they acknowledged that these sources also spread false information. It was felt that misinformation was an important factor preventing or hindering people to get vaccinated.

*"That social media news, that maybe today 100 people got vaccinated and they all died or 3 days after vaccination they died, or they reacted badly or something. Very much impact it had because I delayed getting my vaccine for so long to an extent that I got my vaccination by the time when the government was like, if you don't get vaccinated you will be kicked out of work or school or something." (Clinician, Late Receiver, Matabeleland South)*

**Religion.** According to respondents, religion is critically important for many people in Zimbabwe and religious leaders were actively involved in the vaccination campaign. Almost three quarters of healthcare workers (2,295/3,086, 74%) believed that their community leaders and/or religious leaders would want them to get vaccinated. There was no difference in the proportion of healthcare workers who responded affirmatively to having received the vaccine by age, sex, province, or professional role.

However, despite healthcare workers reporting quantitatively that their religious leaders had no reservations about or were in favor of vaccinations, the qualitative data revealed that some churches did in fact have reservations:

*"For some churches they are in between because they could not reverse what was being said officially, they would say go and get vaccinated but deep-down people would be saying we can't get a vaccine that we don't understand" (Clinician, Late Receiver, Harare).*

Some healthcare workers felt that select church leaders did not encourage vaccination and were rather negative towards COVID-19 vaccines. Participants reported that some churches, specifically the Pentecostal churches and the Apostolic, claimed that their followers "...*will be protected by the Holy Spirit.*" (Clinician, Early Receiver, Harare). One participant said that they had heard church leaders preaching that:

*"...the vaccine is for the triple six, so much so that even up to now, some people have not taken up the vaccine, because they think it's satanism" (Clinician, Late Receiver, Harare).*

Important myths included the risk of death two years after receiving the vaccine, an association between COVID-19 vaccines and satanism, and the perception that COVID-19 vaccines were the *"mark of the beast".* One respondent reported that he had been led to believe that:

*"...you die after 2 years. They were saying the injection has a period of survival just for 2 years. Then you die after 2 years that is what people were saying" (Non-clinician, Early Receiver, Harare).*

**Perceptions of vaccine efficacy and safety.** Generally, healthcare workers perceived the vaccine as 'moderately safe' (1,088/3,086, %) or 'very safe' (1,665/3,086, %). Older HCWs ($\geq$40 years) were more likely to perceive COVID-19 vaccines as very safe (735/1,226, 60%) compared to those <40 years old (929/1,858, 50%).

When probing for detail, healthcare workers voiced some concerns regarding vaccine safety, specifically due to side effects. While none of the interviewed healthcare workers had experienced side effects themselves, they said that some of their friends, colleagues, and family members had experienced symptoms which they believed were due to vaccination.

*"[L]ike there is this nurse that I saw, she had a reaction; she had some sort of funny reaction as if it was like burns. I don't know but she had a reaction, so that on its own is a push factor. She had some complications, and she was admitted, that's a push factor, when people tell you that they have reacted". (Clinician, Early Receiver, Bulawayo)*

The origin (China) of the vaccine was also raised as a cause of concern, specifically because China was the origin of the pandemic and due to theories that the pandemic was man-made.

*"The fact that our vaccine came from China and yet the disease itself, started in China. It seemed like the vaccine coming from China, there are motives to kill us all. That's how it seemed, why did the vaccine come from China?.... From everyone, everyone was just concerned about how people died in China and how the disease started. Then it said they have found a vaccine, yet the disease was from there. There were stories that the disease was man made." (Clinician, Late Receiver, Manicaland).*

Reports of break-through infections further decreased the confidence into the vaccine. Participants questioned the effectiveness of the vaccine:

"*Yes, after being vaccinated. I was talking to one nurse who was saying she wasn't feeling well. I hadn't seen her in a long time, and I asked her where she had gone and she said she had been sick, COVID-19. I asked her if she had been vaccinated and she said she was vaccinated, you could see that she was doubting the vaccine. And there are some who were never vaccinated but up to now they have never been diagnosed of COVID-19*" (Clinician, Late Receiver, Harare).

**SARS-CoV-2 infection an occupational risk.** The risk of severe infection and death was seen as a real possibility and motivation for taking up the vaccine. Healthcare workers felt that they were at a heightened risk of contracting the infection because of the nature of their work. The vaccine was seen as an extra layer of protection (an alternative or additional "personal protective equipment") in situations where there was a breakdown of infection prevention and control because people around them were not adhering to prevention and control measures.

"*The risk of getting COVID-19 here is very high. As individual who works in the outpatient's department. We are the ones that welcome patients. We are the face of the hospital that receive patients even if they do then go to the wards, but patients come through our hands first, whether positive or negative.*" (Clinician, Late Receiver, Manicaland)

"*And then pull factors that thing that you are working with people that are suffering from COVID-19. And you have nothing to protect yourself, you just feel that you have to go and get vaccinated.*" (Clinician, Early Receiver, Bulawayo)

Risk of infection was perceived to be omnipresent. Healthcare workers felt unsafe even away from work because of transmission in the community. What put them at additional risk in the community (over and above other people) was that community members frequently visited their homes to seek health advice (because they were healthcare workers).

People with chronic disease such as HIV and diabetes felt even more vulnerable and hence were anxious to get vaccinated as early as possible.

"*I can easily contract it because of my condition, that I am HIV positive, so we are at risk of contracting a lot of diseases. Because our immune system is weak and is unable to fight strong infections. So that risk makes me afraid that I can contract COVID-19*" (Non-clinician, Early Receiver, Harare).

**Employment and access to services.** Some institutions, especially those run by the government, mandated their employees to be vaccinated. In addition, statements were made that unvaccinated people would not be paid or denied entry into workplaces. Some people accessed vaccination because of work requirements.

"*A lot of people are being pushed [into being vaccinated] by work because a lot of institutions are saying if you are not vaccinated then we won't hire you, so that ends up pushing people.*" (Non-clinician, Early Receiver, Harare).

"*Yes, I have seen people come saying 'I work for a private company. So, they are saying they want everyone vaccinated. If you are not vaccinated, you won't work.*" (Clinician, Early Receiver, Bulawayo).

Also, it was felt that vaccination was mandatory to access various services and institutions. Such institutions included hospitals, churches, and subsidised transport. Some believed that "*nowadays you cannot do anything without being vaccinated,*" (Clinician, Early Receiver, Harare).

"*Some of the push factors are that maybe they say that if you are not vaccinated you will not board the ZUPCO buses or that you won't be able to enter into the supermarket. Or you won't be able to go to the bar or you won't be able to travel from Harare to Bulawayo or Harare to Mutare. Without the vaccination card I feel these are some of the things that will influence people to get vaccinated*" (Non-clinician, Early Receiver, Harare).

**Other people's experiences and recommendation for vaccination.** More cautious participants initially delayed vaccination to observe what would happen to those who were vaccinated. The experience of other co-workers, particularly seniors (managers and supervisors), friends, and relatives and their encouragements had great influence on vaccine uptake.

"*So, I was one of those people that wanted to wait and see what happens to those that have been vaccinated in 5 years. But because seeing that those vaccinated were not having any side effects I just decided to be vaccinated*" (Clinician, Late Receiver, Mashonaland Central).

"*Because most of my peers have been vaccinated and they have been encouraging me to get vaccinated. So, if I make that decision, they will be happy.*" (Clinician, Never Vaccinated, Mashonaland Central).

With vaccines becoming more widely available and the number of vaccinations administered increasing, people were less reluctant to receive their vaccine: "*because they discovered nothing* [wrong] *was going on with them* [the vaccinated]", *(Clinician, Late Receiver, Harare).*
Even though some healthcare workers were initially reluctant and based their decisions to get vaccinate on first observing the outcomes of their vaccinated clients, at the time of the survey a high proportion of healthcare workers (2,855/3,086, 93%) said that they would recommend COVID-19 vaccine to eligible patients. About three quarters of the healthcare workers (2,270/3,086, 74%) thought most of their close family members and friends would want them to get a COVID-19 vaccine. One healthcare worker said:

"*I would also recommend. I think because it didn't give me any problems and maybe the fact that from what I have been hearing that, if you are not vaccinated and if you catch COVID it might be less severe. So, I would recommend.*" (Clinician, Early Receiver, Bulawayo).

**Vaccine availability and access.** According to healthcare workers, provision of vaccine services at health facilities prolonged waiting times for vaccination and other services. Vaccine stocks were sometimes running low and thus people who attended the service for their first shot were turned away as available vaccines were reserved for those attending to receive the second vaccine dose. Though vaccines were eventually delivered to all provinces, initially vaccines were only available in selected approved health facilities in Harare and Bulawayo.

"*It's not always available, sometimes it runs out. Sometimes they say they have run out of first dose; they only have second dose so its vice versa.*" (Clinician, Late Receiver, Harare)

"*There was a time when the first dose was not available, but the second was always there. Some time back at the beginning it was available, but there came a time when it was no longer available.*" (Clinician, Early Receiver, Harare).

However as for healthcare workers themselves, very few reported that the reason for not getting vaccinated were logistical reasons such as long waiting time (2% [n = 2/98]) and vaccine stocks (1% [n = 1]).

## Discussion

In this mixed-method study we explored COVID-19 vaccine uptake among healthcare workers in Zimbabwe. Understanding vaccine uptake and the reasons behind it, as well as remaining concerns that were not adequately addressed, is important for developing new vaccine approaches. With WHO declaring COVID-19 as no longer constituting a global health emergency, vaccine strategies have changed accordingly. Nonetheless, in these new strategies healthcare workers remain a key target population, with the Africa CDC's proposed "100-100-70" targeting the vaccination of 100% of healthcare workers [18]. Achieving these ambitious targets will require careful appraisal of successes, challenges, and opportunities for improved vaccination coverage.

The remarkably high (>97%) prevalence of self-reported COVID-19 vaccine uptake in our study is in stark contrast to other studies from Africa which reported COVID-19 vaccine coverage of 33% and 90% in two settings in Nigeria, respectively, 49% in Somalia, 62% in Ethiopia, 69% in Uganda [32–36]. The main reasons for not being vaccinated in these studies were safety concerns, fear of side effects, and non-availability of the vaccine. A more granular analysis of our results revealed that time of uptake (i.e., whether a person received a vaccine 'early' or 'late') was influenced by geographical location, known chronic conditions, level of education, and professional role. However, overall, there were no clear demographic or behavioural predictors of receiving the vaccine 'early' or 'late' above the patterns of availability and access within the country. For instance, the Bulawayo and Harare provinces, the primary 'hotspots' of the epidemic, received the vaccine first [37], largely due to vaccine supply and availability, which likely influenced who got the vaccine early or late among provinces and hence among the healthcare workers participating in this study. This country-level distribution strategy was in itself shaped by the wider geopolitical situation, notably global vaccine inequality, bilateral arrangement with China, and erratic vaccine supply [38]. Despite high overall rates of vaccination, these inequities and access-related challenges were evidently at play in our research within the Zimbabwean context.

Beyond the broader patterns of access, availability, and distribution, we identified several socio-behavioural influencers that contributed to a comparatively high vaccine uptake as identified by other studies elsewhere [38, 39]. First, many of the healthcare workers felt that the vaccination campaign was effective and paired with well-designed information, education, and communication. Good communication from formal channels helped alleviate concerns and counter misinformation. This perception was not without exception or reservations, however, as one of the main reasons for not getting vaccinated among healthcare workers was pregnancy, suggesting that there were key gaps and inconsistencies in information being provided. Information was also noted as not provided in Zimbabwean local languages, which may not have impacted healthcare workers uptake (who are trained in English) but may have impact on uptake among the general population.

Moreover, concerns about the vaccine were certainly present among healthcare workers. Participants expressed concerns about the vaccine's origin, the speed taken to develop which led to safety and efficacy concerns. These concerns were clearly amplified by rumours and misinformation that were rife during the pandemic, as highlighted by many of our participants. Studies elsewhere have shown the influence of COVID-19 misinformation on public confidence, leading to vaccine scepticism [39, 40]. In our study, social media and religious leaders

were considered to be important opinion leaders and spreaders of (mis)information among the community. The wide influence of religious leaders highlights the importance of engaging them in health-related matters and specifically on vaccinations. While our findings show that misinformation did not hugely affect the vaccine uptake among healthcare workers as in other settings [39, 41], these concerns need to be taken seriously in future risk communication and community engagement (RCCE) strategies. Healthcare workers are considered a source of trusted information for the community and hence gatekeepers to high vaccination coverage of the general population.

A second main reason behind high uptake in our study is explained by healthcare workers' occupational risk perception. In a study which examined the relationship between risk perception, vaccine trust, and vaccine uptake the authors found that perceived risk of SARS-CoV-2 infection increased COVID-19 vaccination by 1.6 times [42]. In our study, healthcare workers felt vulnerable to SARS-CoV-2 infections and because they believed that vaccines were generally safe, they then got vaccinated. In the absence of adequate personal protective equipment, healthcare workers viewed COVID-19 vaccination as an extra layer of protection. Generally, those who were objectively at higher risk of severe disease (such as older people, people with obesity and/or co-morbidities) were more likely to be vaccinated earlier. This may be due to their own perceived higher risk, or it may be due to initial prioritisation of these at-risk groups among healthcare workers and also an effect of the RCCE campaign.

Finally, and perhaps most influential of all, mandatory vaccination policies, though not stringently applied, influenced the vaccine uptake among healthcare workers in Zimbabwe. 'Mandatory vaccination' is defined as 'a policy that establishes a requirement that an individual be vaccinated based on their status or their eligibility to access societal or governmental benefits' [43]. Mandatory vaccination can be effective in increasing uptake as found in other settings such as Somalia, and Uganda [33, 34]. However, such a strategy has the potential to undermine trust of both the vaccine but also the authority mandating vaccination. While stringent lockdowns and mandatory vaccination may have been acceptable during a time of intense COVID-19 transmission, high infection and case-fatality rates, such measures are unlikely to be acceptable in the post COVID-19 emergency era. Studies on COVID-19 vaccine mandatory have shown resistant perception about mandating the vaccine [44, 45]. Mandatory vaccination most certainly does not solve often quite legitimate concerns people may have about new vaccines and treatment but may exacerbate them [46]. Therefore, mandating vaccination needs to be carefully balanced with other interventions. It might be noted that historically, Zimbabwe achieved high vaccine uptake across different vaccination programs including Expanded Program on Immunization for children, typhoid conjugate vaccine (84.5% for children and adults and human papilloma virus vaccine (88–94%) for young adults [47, 48]. Mandatory vaccine policies were not in place for these other vaccine-preventable diseases.

As our study and several others have shown [49–52], healthcare workers are not simply passive participants of a vaccination campaign, as seen by their calculated risk assessments and astute socio-political observations. Given their centrality to current and future vaccine policy, their active involvement in the development of strategies is key, especially in addressing legitimate concerns which can be packaged in RCCE activities. Increasing vaccine choices and transparency on adverse effects after immunization through surveillance is likely to address healthcare workers' legitimate concerns and anxieties. Beyond healthcare workers, misinformation disseminated on social media should be addressed proactively by working with influencers such as religious leaders and public figures.

The strengths of this study lie in the use of a mixed-method approach investigating self-reported vaccine uptake and associated reasons. Our use of both qualitative and quantitative methods allowed us to combine the strengths of deductive and inductive reasoning, while

offsetting the limitations of using exclusively one approach. This complementary approach enabled us to produce a robust and reliable, but also nuanced and contextual understanding of vaccine uptake among healthcare workers in Zimbabwe. The sample size was large both with regards to the number of healthcare workers included and the number of health facilities, with the latter including a diverse range of health facilities from tertiary to primary level and across different provinces. The limitations include that vaccination was self-reported and not verified by checking vaccination cards and thus may have been subject to social desirability (by which healthcare workers could have responded in a generally positive way which conforms to their profession) and/or recall bias (specifically the date of vaccination). Also, healthcare workers were self-selected from those who came forward to access the health check-up service. This may have introduced selection bias as those healthcare workers who took up the health check service may have been more health conscious and thus more likely to be vaccinated.

## Conclusion

In conclusion, vaccine uptake among healthcare workers in Zimbabwe is high despite the limited vaccine choices, misinformation, hesitancy, and health systems challenges. The key factors positively affecting uptake were a generally well organised information and communication campaign (with certain limitations) and occupational risk perception coupled with 'mandatory vaccination'. (Mis)information on social media and through religious leaders as well as vaccine-related logistics were also thought to be important. Active engagement of healthcare workers in vaccine strategy is crucial for understanding current concerns and for developing context-sensitive strategies that address remaining concerns of healthcare workers and wider population.

## Supporting information

**S1 Checklist. PGPH-D-23-01351 inclusivity questionnaire.** The form has been filled in as per the Plos Global Health inclusivity requirements.
(DOCX)

**S1 Table. Multivariable logistic regression investigating the association between client characteristics and receiving the vaccine late or never receiving the vaccine (n = 1,551).** The table illustrates the results of a multivariable logistic regression sensitivity analysis. The p-value was derived using a likelihood ratio test and the abbreviations: aOR: adjusted odds ratio, 95% CI: 95% confidence interval.
(DOCX)

**S2 Table. Healthcare workers COVID-19 vaccine uptake emerging themes, codes, and additional quotes.** The table has three columns showing themes, codes, and relevant supporting quotes that emerged from the qualitative thematic data analysis.
(DOCX)

## Author Contributions

**Conceptualization:** Tinotenda Taruvinga, Justin Dixon, Katharina Kranzer.

**Data curation:** Tinotenda Taruvinga, Leyla Larsson, Nicole Redzo, Justin Dixon, Katharina Kranzer.

**Formal analysis:** Tinotenda Taruvinga, Rudo S. Chingono, Leyla Larsson, Justin Dixon, Katharina Kranzer.

**Funding acquisition:** Rashida A. Ferrand, Katharina Kranzer.

**Investigation:** Tinotenda Taruvinga, Rudo S. Chingono, Edson Marambire, Sibusisiwe Sibanda, Farirai Nzvere.

**Methodology:** Tinotenda Taruvinga, Farirai Nzvere, Justin Dixon, Katharina Kranzer.

**Project administration:** Tinotenda Taruvinga.

**Resources:** Katharina Kranzer.

**Supervision:** Rudo S. Chingono, Edson Marambire, Agnes Mahomva, Rashida A. Ferrand, Justin Dixon, Katharina Kranzer.

**Validation:** Tinotenda Taruvinga, Rudo S. Chingono, Chiratidzo E. Ndhlovu, Hilda Mujuru, Edwin Sibanda, Prosper Chonzi, Maphios Siamuchembu, Rudo Chikodzore, Agnes Mahomva, Rashida A. Ferrand, Justin Dixon, Katharina Kranzer.

**Visualization:** Tinotenda Taruvinga, Leyla Larsson, Ioana D. Olaru.

**Writing – original draft:** Tinotenda Taruvinga, Justin Dixon, Katharina Kranzer.

**Writing – review & editing:** Tinotenda Taruvinga, Edson Marambire, Leyla Larsson, Ioana D. Olaru, Sibusisiwe Sibanda, Farirai Nzvere, Nicole Redzo, Chiratidzo E. Ndhlovu, Simbarashe Rusakaniko, Hilda Mujuru, Edwin Sibanda, Prosper Chonzi, Maphios Siamuchembu, Rudo Chikodzore, Agnes Mahomva, Rashida A. Ferrand, Justin Dixon, Katharina Kranzer.

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
