## [Decision Letter · Decision Letter 0]

21 Aug 2023

PGPH-D-23-01351

Exploring COVID-19 vaccine uptake among healthcare workers in Zimbabwe:  A mixed methods study

Dear Dr. Taruvinga,

Thank you for submitting your manuscript to PLOS Global Public Health. After careful consideration, we feel that it has merit but does not fully meet PLOS Global Public Health’s publication criteria as it currently stands. Therefore, we invite you to submit a revised version of the manuscript that addresses the points raised during the review process.

We look forward to receiving your revised manuscript.

Kind regards,

Oghenebrume Wariri, MD

Academic Editor

Journal Requirements:

Additional Editor Comments (if provided):

Dear Authors,

Kindly ensure that you address ALL the issues raised by Reviewers 1 and 2 by providing a point-by-point response on how you addressed the issues in the manuscript. Please pay attention to the issues related to enhancing the quantitative component of the manuscript. Ensure you include supplementary statistical tests (e.g., binary associations, logistic regression models, etc.) to elaborate the connections between the independent variables and the dependent variable, specifically categorizing healthcare workers into "early" and "late" vaccine recipients. Strengthening the quantitative analysis will provide a better understanding of the factors influencing vaccine uptake among healthcare workers, thus contributing to the robustness of the study's findings.

Reviewers' comments:

Reviewer's Responses to Questions

**Comments to the Author**

1. Does this manuscript meet PLOS Global Public Health’s publication criteria? Is the manuscript technically sound, and do the data support the conclusions? The manuscript must describe methodologically and ethically rigorous research with conclusions that are appropriately drawn based on the data presented.

Reviewer #1: Yes

Reviewer #2: Yes

2. Has the statistical analysis been performed appropriately and rigorously?

Reviewer #1: No

Reviewer #2: I don't know

3. Have the authors made all data underlying the findings in their manuscript fully available (please refer to the Data Availability Statement at the start of the manuscript PDF file)?

Reviewer #1: Yes

Reviewer #2: Yes

4. Is the manuscript presented in an intelligible fashion and written in standard English?

Reviewer #1: Yes

Reviewer #2: Yes

5. Review Comments to the Author

Reviewer #1: I appreciate the opportunity to review your mixed-method study aiming to understand vaccine uptake, perceptions, and attitudes among healthcare workers in Zimbabwe. Below are my comments and suggestions that I believe would enhance the quality and impact of your research.

Major Points

Introduction:

• It is imperative for the authors to cite and discuss prior research studies addressing the same research problem. This context will underscore the novelty of your study and help differentiate it from previous attempts.

• Clearly state the aim and objectives of your study at the conclusion of the introduction section to provide a concise overview of your research goals.

Methods:

• In describing your study design as mixed-method, it is crucial to bolster the quantitative component of your manuscript. Authors should employ additional statistical tests (e.g., binary associations, logistic regression models, etc.) to describe and explain the associations between the independent variables and the dependent variable (healthcare workers categorized as receiving the vaccine “early” and those categorized as “late”).

• Authors should provide more information regarding the study population (e.g., what kind of healthcare professionals, students, adults, working facilities, etc.).

• The sampling strategy should be clearly defined and reported in the methods section.

• Authors should clarify how the sample size was determined.

• The authors fail to provide any information in their manuscript that would allow readers to understand how the sample differs from the target population, particularly in terms of key sociodemographic characteristics such as age, gender, and others. This lack of information raises concerns about the representativeness of the sample.

• No information is given regarding a pilot study for testing the survey tools (e.g., questionnaire, interviews, etc.).

• The authors did not provide any information regarding the methods used to collect the data (e.g., questionnaires). Detailed information (e.g., content, type, and number of questions, etc.) on this is crucial for the study’s validity.

• Authors should clarify how they estimated the reliability or internal consistency of the questionnaire used, using, for example, Cronbach’s alpha to measure whether a score is reliable.

• More information should be provided regarding the considerations concerning the literature review (please add the references) and the authors' previous research experience in developing the questionnaire.

• Lines 147-153: Authors need to provide more information and reasoning on how they selected the healthcare professionals that were included in the qualitative study.

• Detailed information should be added about the methods and procedures of the Qualitative Data Analysis (thematic analysis).

• The study should use additional statistical analysis. Specifically, information on how the authors assessed the normality of numeric variables and any relevant assumptions made for the chosen statistical tests or models should be reported to enhance the transparency and reliability of the statistical analysis. Also, information should be added regarding the logistic regression analysis used (e.g., independent and dependent variables, assumptions, etc.)..

Results:

• Detailed findings from the statistical analyses used such as the logistic regression models should be reported in the text of the Results section. The authors should create new tables reporting the results from these analyses, which should be included among the main tables of the manuscript.

• Themes and categories identified from qualitative data analysis should be reported in a corresponding Table.

Discussion:

• Authors should compare their findings not only with studies conducted in Zimbabwe but also with other studies.

• The discussion section should explore the underlying meaning of the research and its potential implications in other areas of study. It would be beneficial for the authors to discuss how their findings contribute to the existing knowledge and highlight the significance of their research within a broader context. Additionally, they should consider addressing possible improvements or future directions that can further develop the concerns of their research.

• Additional limitations should be added in the limitations section.

• Line 473: It is necessary to include a discussion of the levels of support for a mandatory vaccination policy among HCPs in Zimbabwe and other countries. For example, https://doi.org/10.1016/j.idnow.2021.08.004, https://doi.org/10.3390/ijerph18136688, https://doi.org./10.3389/fpubh.2022.897526 , https://doi.org/10.3390/vaccines9060580 , https://doi.org/10.3390/vaccines9080889 , https://doi.org/10.3390/hygiene3030021 ) and other articles could be added.

General Points:

• Tables should be reviewed for typos.

• The authors could go through the manuscript once more to correct language errors.

Reviewer #2: 

The authors should consider addressing the issues raised below to improve the overall quality of the manuscript

Line 73 Their roll-out has reduced morbidity, severity and deaths [1–3].

Comment: The sentence is a bit short kindly provide more detail.

Line 75 and 77

Comment: What is vaccine nationalism? Define it.

Line 81 For instance Zimbabwe, the focus of our research, did not initially sign up…

Comment: Re-read and edit this sentence slightly

Line 93 …high-income and LMICs suggest…

Comment: Write high-income countries instead

Line 100 …priority group with a target set to reach 100% COVID-19 vaccine coverage among this group [18].

Comment: Reach them by which year?

Line 102 …vaccination strategies were successful…

Comment: Which past vaccination strategies? All of them or COVID-19 related?

Line 110: Data was collected as part of a study providing a comprehensive health check to healthcare workers in…

Comment: Would be clearer to say data was collected as part of another study.

Line 112: …interviews were conducted with selected healthcare workers accessing the service between June 2021…

Comment: Accessing what service?

Line 126: The country approved the use of Sinopharm and Sinovac…

Comment: You mean Sinopharm and Sinovac vaccines.

Line 124 Zimbabwe’s COVID-19 vaccination campaign

Comment: Very nicely written section on the history and flow of events

Line 137 …third wave primarily driven by the delta variant which resulted in the highest number of deaths…

Comment: Be more specific. You mean third wave of the COVID-19 pandemic. What is the delta variant?

Line 143 Procedures

Comment: An overall comment about this section is to sign post where you switch from speaking about a qualitative in-depth interview to a quantitative questionnaire/survey. E.g. on line 151 write “For the qualitative in-depth interviews…” to match with line 159 where you state “…as part of the quantitative questionnaire…”

Line 144 Following verbal informed consent and prior to accessing the health check…

Comment: What is the health check? Good to detail here at first instance as you go on to mention it further down in this section.

Line 146 and 147 Questions about perceived vaccine safety were based on a WHO survey tool [18].

Comment: Where did you get guidance questions about vaccine uptake?

Line 147 and 148 Written informed consent from selected healthcare workers was obtained for in-depth interviews…

Comment: State how you got informed consent for those who took the questionnaire.

Lines 147 to 149 Written informed consent from selected healthcare workers was obtained for in-depth interviews to better understand healthcare workers reasons for taking or not taking up the COVID-19 vaccine and the challenges they may have faced in the process.

Comment: Best to break up this sentence. First, you mean to say that written informed consent was obtained from all selected healthcare workers. Second, you mean to say that in-depth interviews were used to better understand healthcare workers reasons for taking or not taking up the COVID-19 vaccine and the challenges they may have faced in the process. I recommend you give a citation here on the use of in-depth interviews in research.

Line 151 For the in-depth interviews, healthcare workers were purposively selected…

Comment: Briefly define what it means to purposively select participants and provide citation.

Line 153 A total of 17 in-depth interviews were conducted, after reaching a data saturation point [29].

Comment: Expand briefly on the meaning of data saturation.

Line 162 The interviews were conducted in English, Ndebele, or Shona according…

Comment: Do you mean both the interviews and questionnaires?

Line 164 …all possible probing questions.

Comment: Define what probing questions are and why you needed to use them.

Line 173 …of June 2021, while somebody receiving the first vaccine dose after June 2021 was categorised as “late”.

Comment: Instead of “somebody” write “those”

Line 188 In-depth interviews were audio-recorded, transcribed, and translated. During the interview, research…

Comment: Transcribed and translated by who?

Line 191 …software NVivo 12, which was used to perform thematic analysis. Thematic analysis was performed on…

Comment: Describe in more detail what thematic analysis is and how you did it.

Line 195 Quantitative findings were triangulated with the themes emerging from the qualitative data throughout the…

Comment: Good starting explanation of mixed methods data analysis but need to detail more.

Line 203 …waived the necessity for a written informed consent to facilitate access to the service.

Comment: How would written informed consent have been a deterrent to access to the service?

Lines 438-447 Beyond the broader patterns of access, availability, and distribution, we identified several socio behavioural influencers that contributed to a comparatively high vaccine uptake.

Comment: Are these socio behavioural influencers also found in published literature?

Line 498 The strengths of this study lie in the use of a mixed-method approach investigating self-reported vaccine…

Comment: Add one more sentence on why mixed -method approach is a strength.

Line 503 …subject to social desirability and/or recall bias (specifically the date of vaccination). Also, healthcare…

Comment: Define social desirability

Line 505 …may have introduced selection bias as those healthcare workers who took up the health check service…

Comment: Define selection bias

6. PLOS authors have the option to publish the peer review history of their article (what does this mean?). If published, this will include your full peer review and any attached files.

**Do you want your identity to be public for this peer review?** For information about this choice, including consent withdrawal, please see our Privacy Policy.

Reviewer #1: No

Reviewer #2: **Yes: **Dr. Penda Johm

---

## [Decision Letter · Decision Letter 1]

3 Nov 2023

Exploring COVID-19 vaccine uptake among healthcare workers in Zimbabwe:  A mixed methods study

PGPH-D-23-01351R1

Dear Mr Taruvinga,

We are pleased to inform you that your manuscript 'Exploring COVID-19 vaccine uptake among healthcare workers in Zimbabwe:  A mixed methods study' has been provisionally accepted for publication in PLOS Global Public Health.

Best regards,

Oghenebrume Wariri, MD

Academic Editor

We are pleased that you incorporated the feedback provided by the reviewers into your revised manuscript and provided a point-by-point response to their comments. I wish your team well in your future research endeavours.

Reviewer Comments (if any, and for reference):

Reviewer's Responses to Questions

**Comments to the Author**

1. If the authors have adequately addressed your comments raised in a previous round of review and you feel that this manuscript is now acceptable for publication, you may indicate that here to bypass the “Comments to the Author” section, enter your conflict of interest statement in the “Confidential to Editor” section, and submit your "Accept" recommendation.

Reviewer #2: All comments have been addressed

2. Does this manuscript meet PLOS Global Public Health’s publication criteria? Is the manuscript technically sound, and do the data support the conclusions? The manuscript must describe methodologically and ethically rigorous research with conclusions that are appropriately drawn based on the data presented.

Reviewer #2: Yes

3. Has the statistical analysis been performed appropriately and rigorously?

Reviewer #2: I don't know

4. Have the authors made all data underlying the findings in their manuscript fully available (please refer to the Data Availability Statement at the start of the manuscript PDF file)?

Reviewer #2: Yes

5. Is the manuscript presented in an intelligible fashion and written in standard English?

Reviewer #2: Yes

6. Review Comments to the Author

Reviewer #2: You addressed all my comments but please make sure to address those of anonymous reviewer #1, especially taking time to detail your methods and findings from the statistical analyses. I also agree that it would read better if you instead reported the themes and categories identified from qualitative data analysis in a table as the quotes are making the results section longer.

7. PLOS authors have the option to publish the peer review history of their article (what does this mean?). If published, this will include your full peer review and any attached files.

**Do you want your identity to be public for this peer review?** For information about this choice, including consent withdrawal, please see our Privacy Policy.

Reviewer #2: **Yes: **Dr. Penda Johm
